# HMGA1 Modulates Gene Transcription Sustaining a Tumor Signalling Pathway Acting on the Epigenetic Status of Triple-Negative Breast Cancer Cells

**DOI:** 10.3390/cancers11081105

**Published:** 2019-08-02

**Authors:** Carlotta Penzo, Laura Arnoldo, Silvia Pegoraro, Sara Petrosino, Gloria Ros, Rossella Zanin, Jacek R. Wiśniewski, Guidalberto Manfioletti, Riccardo Sgarra

**Affiliations:** 1Department of Life Sciences, University of Trieste, 34127 Trieste, Italy; 2Department of Proteomics and Signal Transduction, Max Planck Institute of Biochemistry, 82152 Martinsried, Germany

**Keywords:** High Mobility Group A, breast cancer, TNBC, epigenetic, RSK2, CBP, histone H3, histone H2B

## Abstract

Chromatin accessibility plays a critical factor in regulating gene expression in cancer cells. Several factors, including the High Mobility Group A (HMGA) family members, are known to participate directly in chromatin relaxation and transcriptional activation. The HMGA1 oncogene encodes an architectural chromatin transcription factor that alters DNA structure and interacts with transcription factors favouring their landing onto transcription regulatory sequences. Here, we provide evidence of an additional mechanism exploited by HMGA1 to modulate transcription. We demonstrate that, in a triple-negative breast cancer cellular model, HMGA1 sustains the action of epigenetic modifiers and in particular it positively influences both histone H3S10 phosphorylation by ribosomal protein S6 kinase alpha-3 (RSK2) and histone H2BK5 acetylation by CREB-binding protein (CBP). HMGA1, RSK2, and CBP control the expression of a set of genes involved in tumor progression and epithelial to mesenchymal transition. These results suggest that HMGA1 has an effect on the epigenetic status of cancer cells and that it could be exploited as a responsiveness predictor for epigenetic therapies in triple-negative breast cancers.

## 1. Introduction

Most of the chemical, biological, and mechanical stimuli a cell is subjected to, such as for instance hormones, growth factors, cytokines, cell-cell or cell-extracellular matrix contacts, are conveyed by signalling pathways to chromatin, leading to the activation of specific transcriptional programs. In cancer cells, these signalling networks are perturbed by the misregulation of key components, which are responsible for improper responses that constitute oncogenic boosts leading to the acquisition of the typical cancer aggressive traits, i.e., the so called cancer hallmarks [1]. A rewiring of gene expression is fundamental to acquire these capabilities [1,2]. Chromatin structure poses a steric hindrance to the free access of transcriptional factors/co-activators to DNA and it is overcome by means of epigenetic mechanisms, which finally lead to chromatin opening [3]. The different conformations chromatin can assume are intimately linked to the post-translational modifications (PTMs) of the histone tails protruding from the nucleosomes, the primary packaging scaffold of DNA. Indeed, the different combinations of PTMs of histones constitute a code for the selective binding of a plethora of different proteins, which finally dictate the accessibility of DNA [4].

The High Mobility Group A (HMGA) chromatin architectural factors are DNA–binding proteins encoded by two different genes: HMGA1 [5], which gives rise to two splicing variants (HMGA1a and HMGA1b) [6], and HMGA2 [7], which encodes for the HMGA2 protein (for simplicity, we will refer to HMGA1a and HMGA1b jointly as HMGA1). They have been defined as oncofetal proteins because of their high expression levels in both embryonic and cancer cells [8,9]. These proteins have been intimately associated with the neoplastic transformation process [10,11], and their causal role in cancer development has been firmly established both in vitro and in vivo [12,13,14,15].

HMGA proteins participate, by means of protein/DNA and protein/protein interactions, in the assembly of nuclear macromolecular complexes, i.e., enhanceosomes, at the level of gene regulatory regions [16,17] and modulate, by protein/protein interactions, the activity of important cellular regulators [18,19,20]. Indeed, HMGA1 is able to inhibit the proapoptotic activity of p53 in thyroid cancer cells [18] while both HMGA1 and HMGA2 can enhance transcription factor E2F1 (E2F-1) activity displacing histone deacetylase 1 (HDAC1) from retinoblastoma-associated protein (pRb) and enhancing E2F-1 activity thus overcoming pRb mediated G0 arrest and promoting the development of pituitary adenomas [19,20]. HMGA proteins are among the most connected proteins in the nuclear environment of transformed cells [21,22], and it is therefore not surprising that they are considered key nodes in the chromatin network [23].

HMGA1 protein contacts core histones via their protruding tails in the nucleosome context [24]. Given the chromatin architectural role of HMGA proteins, the HMGA/histone interaction was perceived as very important, especially with respect to the action of HMGA1 in altering the positioning/phasing of nucleosomes on specific promoter/enhancer DNA regulatory sequences [24]. Surprisingly, this very interesting aspect regarding the HMGA involvement in chromatin structure and dynamics was not fully explored.

Several groups highlighted a prominent role of HMGA1 in breast cancer (BC) development and progression (reviewed in [25]), showing its major contribution to cell motility [26,27,28,29], stemness and self-renewal [29], and epithelial to mesenchymal transition (EMT) [29,30]. Moreover, HMGA1 was found to promote breast tumorigenesis by affecting the Ras-extracellular signal-related kinase (Ras/ERK) mitogenic signalling pathway [31], DNA repair mechanisms [32], alternative splicing of estrogen receptor [33], breast cell secretome [34], and nuclear stiffness [35]. In addition, HMGA1 was found out to act as a signalling molecule in the extracellular environment promoting invasion and metastasis [36].

In this work, we investigated the relationship between HMGA1 and histones to understand whether HMGA1 could promote BC progression through the modulation of epigenetic mechanisms. 

## 2. Results

### 2.1. HMGA1 Expression Influences the Histone Code

Both in vitro and in vivo studies have highlighted the interaction between HMGA1 and histones H2A, H2B and H3, pointing out the involvement of amino-terminal histone tails in these contacts [24]. HMGA proteins regulate both wide-range and locus-specific chromatin processes, both by shaping DNA and by interacting with several other nuclear partners [23,37]. We therefore hypothesized that physical contact of HMGA1 with core histones could provide the opportunity to affect PTMs of core histone tails, i.e., the histone epigenetic code [4], thus potentially influencing epigenetic events. To this purpose, HMGA1 expression was downregulated by small interfering RNA (siRNA) in MDA-MB-231 cells, where it is expressed at high levels, and samples were analysed for PTMs associated with core histone H3 and H2B tails that have cumulative effects on transcriptional regulation. In particular, we investigated marks of transcriptional activation characteristics of the H3 tail, i.e., S10 and S28 phosphorylation (ph) [38,39,40] and K14 acetylation (ac) [41], and others belonging to H2B, such as K5ac and K16ac [42,43]. Conversely, we checked the levels of methylated histone H3, particularly H3K9me3 and H3K27me3, which are known repressive markers [44,45,46].

As shown in Figure 1A, HMGA1 downregulation clearly impacted the levels of active chromatin marks of both H3 (left side) and H2B (right side), whereas repressive marks seemed to be largely unchanged. In particular, phosphorylation of H3 and acetylation of H2B were the most affected (see Figure 1B for a quantitative evaluation of western blot analyses shown in Figure 1A). We confirmed the modulation of H3S10ph using a different HMGA1–targeting siRNA, different triple-negative breast cancer (TNBC) cellular models (MDA-MB-157 and MDA-MB-468 cells), and a different α–H3S10ph antibody (Appendix A).

The downregulation of both H3S10ph and H2BK5ac was then validated by immunofluorescence (IF) analyses performed on MDA-MB-231 cells (Appendix A). Moreover, both PTMs appeared to be present in the entire cell population, thus suggesting that, in accordance with literature data, they occur mostly in interphasic cells [38,39,40,42,43].

### 2.2. RSK2 Kinase Phosphorylates H3S10 and H3S28 in Interphasic MDA-MB–231 Cells

We initially focused on elucidating the HMGA1-dependent pathway(s) involved in H3S10 and H3S28 phosphorylation. The histone H3 tail is the substrate of many kinases that activate different transcriptional programs in response to several stimuli. We concentrated on the mitogen-activated protein kinase (MAPK) pathways since they are responsible for H3 phosphorylation in interphase cells [47,48]. Downstream of these pathways, mitogen- and stress-activated kinase 1 (MSK1) and ribosomal protein S6 kinase alpha-3 (RSK2) are the proteins principally involved in H3 phosphorylation [47,49]. The Ras-extracellular signal-related kinase (Ras/ERK) mitogenic signalling pathway can activate both MAPK-activated kinases, whereas MSK1/2 can also be activated by p38 under stress conditions or inflammatory cytokines [49,50]. To determine which MAPK pathway was responsible for H3 phosphorylation in our cellular model, we treated MDA-MB-231 cells with two inhibitors, U0126 and BIRB796, which block MEK1/2- and p38-dependent phosphorylation cascades, respectively [50,51]. As shown in Figure 2A, interphasic H3S10ph and H3S28ph were strongly dependent on the activity of MEK1/2 (U0126) and were almost unaffected by p38 inhibition (BIRB796). Moreover, U0126 treatment blocked the phosphorylation of ERK by MEK1/2 without affecting HMGA1 levels (Figure 2B).

Since we demonstrated that interphasic H3S10ph and H3S28ph were mediated by the Ras/Raf/MEK/ERK pathway in MDA-MB-231 cells (Figure 2A), we tested whether ERK activation was dependent on HMGA1 expression. Therefore, we silenced the expression of HMGA1 and analysed ERK1/2 activation. Results clearly indicate that HMGA1 silencing (Figure 2C) did not alter ERK expression or ERK phosphorylation, suggesting that HMGA1 acts downstream of ERK activation. Moreover, to exclude that HMGA1 could affect the expression of the two main downstream kinases responsible for H3 phosphorylation upon ERK pathway activation, i.e., MSK1/2 [47] and RSK2 [49], the same extracts from MDA-MB-231 cells silenced for HMGA1 expression, were analysed for MSK1 and RSK2 expression. The results show that HMGA1 silencing substantially affected neither MSK1 nor RSK2 protein expression (Figure 2C).

To determine the contribution of MAPK-activated protein kinases to the interphasic phosphorylation of H3S10 and H3S28, we treated MDA-MB-231 cells with specific inhibitors of MSK1/2 (SB747651A, [52]) and RSK isoforms (BI-D1870, [53]). In contrast to MSK1/2, RSK isoforms inhibition drastically impaired both H3 phosphorylations (Figure 2D). The H3S10ph decrease was observed within few hours of inhibitor treatment, suggesting that these kinases have a direct role in histone phosphorylation rather than being the result of an indirect effect due to reduced proliferation (Appendix A). RSK2 is one of the main kinases able to phosphorylate histone H3S10 [49] and its specific involvement in this context was confirmed by comparing the silencing of MSK1/2 and RSK2 (Appendix A). A summary of all the inhibitors used and their targets within the studied pathways is shown in Figure 2E.

### 2.3. RSK2 Impairment Mimics the Effects of HMGA1 Silencing

To confirm that interphasic H3 phosphorylation is necessary for gene expression activation, we investigated whether the silencing, or inhibition, of RSK2 affects a subset of genes (Aurora kinase B, AURKB; Kinesin-like protein KIF23, KIF23; Chromosome-associated kinesin KIF4A, KIF4A; Centromere protein F, CENPF) belonging to the HMGA1 signature that we previously identified in BC cells [29] and from now on referred to as the HMGA1 subset signature (Hsss). Therefore, MDA-MB-231 cells were collected after RSK2 silencing or upon treatment with BI-D1870, and the expression of these genes was analysed (Figure 3). 

The expression of Hsss was significantly downregulated upon RSK2 silencing (Figure 3B) and BI-D1870 treatment (Figure 3C). As a control, the expression of Hsss was analysed upon HMGA1 silencing (Figure 3A) together with another set of genes whose expression was not altered by HMGA1 (Appendix A). To confirm that the specificity of gene expression activation is due to RSK2-dependent phosphorylation of H3, the Hsss was analysed upon MSK1/2 silencing and no effect on its level of expression was observed (Appendix A). The dependence of Hsss by HMGA1 and RSK2 was confirmed in MDA-MB-157 cells (Appendix A).

We reported a drastic phenotypic change, the so-called mesenchymal to epithelial transition (MET), as a consequence of the downregulation of HMGA1 expression in mesenchymal-like TNBC MDA-MB–231 cells. This change was associated with reduced cell motility and invasiveness [29]. We therefore hypothesized that RSK2-dependent histone H3 phosphorylation could mediate the activation of a transcriptional program involved in MET. Indeed, MDA-MB-231 cells treated with BI–D1870 underwent substantial changes in their aspect (Figure 4A): from a spindle-shaped morphology to a regular and flattened one, reflecting a typical MET very close to the one obtained by HMGA1 downregulation [29]. In addition, the change in cell morphology occurred very early after BI–D1870 treatment (4 h, Appendix A). BI-D1870 treatment slowed down cell proliferation, as evidenced by a [3-(4,5-dimethylthiazol-2-yl)-5-(3-carboxymethoxyphenyl)-2-(4-sulfophenyl)-2H-tetrazolium]-based (MTS) assay (Figure 4B) and, in addition, it significantly impaired cell migration as evidenced both by wound–healing (Figure 4C,D) and transwell assay (Figure 4E,F). 

Moreover, qRT-PCR analyses showed the down regulation of a set of mesenchymal markers (vimentin, VIM; zinc finger protein SNAI2, SNAI2, and lymphoid enhancer-binding factor 1, LEF1) (Figure 4G), thus supporting a MET. Treatment of another TNBC cell line, i.e., MDA-MB-157, with BI-D1870 reproduced the same effects observed in MDA-MB-231; in particular, albeit at a lower extent, we observed the same morphological transition, a decrease in cell proliferation, and the impairment of cell motility (Appendix A). Overall, these data support that RSK2-dependent interphasic phosphorylation of H3 plays a crucial role in the activation of an HMGA1-driven transcriptional program that is associated with cell aggressiveness.

HMGA1 is known to be involved in the modulation of the expression of several genes by directly binding to cis regulatory regions. We chose three different genes that are part of the HMGA1 signature [29] and that we demonstrated to be transcriptionally regulated by HMGA1: G1/S-specific cyclin E2 (CCNE2) [54], plasminogen activator inhibitor 1 (SERPINE1) [34], and aurora kinase B (AURKB) [29]. By using the RSKs inhibitor BI–D1870, whose effects closely resemble the gene transcriptional and biological effects achieved with HMGA1 silencing, we demonstrated that H3S10ph is specifically decreased in several regulatory regions of these genes (Figure 5), thus suggesting a chromatin–related involvement of this epigenetic mark in gene expression regulation.

### 2.4. CBP Impairment Mimics the Effects of HMGA1 Silencing

After establishing the pathway involved in HMGA1–dependent phosphorylation of H3, we moved to H2B. Histone H2B acetylation has been extensively linked to the activity of two distinct histone acetyltransferases (HATs), CREB-binding protein (CBP) and its paralogue p300 (E1A binding protein p300), which display strong sequence similarities and functional homologies and could either play overlapping or unique functions [55,56]. We investigated whether the acetylation status of H2B in MDA-MB-231 cells was dependent on CBP/p300 activity (Figure 6). 

For this purpose, we used the pyrazolone-containing small molecule inhibitor C646 [57]. Considering that C646 competes with acetyl-CoA for CBP/p300 binding [58], cells were serum-starved for 24 h prior to C646 treatment to lower acetyl coenzyme A (acetyl-CoA) levels and increase the efficiency of C646 inhibition. Cells were treated with C646 or DMSO as a vehicle control and analysed for H2BK5ac. Figure 6A shows a notable decrease in the levels of H2BK5 acetylation, while total H2B protein levels remained almost unchanged (Figure 6A, left panel). 

Inhibition of CBP/p300 concomitantly led to the downregulation of Hsss in MDA-MB-231 cells (Figure 6A, right panel), thus suggesting that the CBP/p300-mediated acetylation of H2B could be involved in the activation of HMGA1-linked transcriptional programs.

Despite their high degree of homology, CBP and p300 also have non-redundant in vivo functions. To dissect the roles played by these two acetyltransferases within the HMGA1–dependent epigenetic reprogramming in BC cells, we silenced the expression of CBP and p300 independently and examined both H2B acetylation levels and the expression of Hsss. When the expression of CBP (Figure 6B, left panel) and p300 (Figure 6C, left panel) was silenced in MDA-MB-231 cells, there were a significant downregulation of H2BK5ac levels compared to control cells. Interestingly, Hsss was found to be significantly downregulated only upon CBP silencing (Figure 6B, right panel), whereas its expression was not altered in p300-silenced cells (Figure 6C, right panel). These data suggest that H2B acetylation relies both on the activity of CBP and p300 in MDA-MB-231 cells, but only CBP is involved in the gene expression regulation of Hsss. The dependence of Hsss by CBP was confirmed in MDA-MB-157 cells (Appendix A).

### 2.5. Histone H3S10ph and Histone H2BK5ac are Interdependent PTMs

Histone acetylation and phosphorylation have been coupled to transcriptional activation. CBP has been shown to interact with unphosphorylated RSK2, forming a complex where both CBP and RSK2 are kept inactivated. Upon mitogenic stimuli leading to the phosphorylation of RSK2 at S227, RSK2 and CBP dissociate and drive gene expression activation by acting as a histone kinase and histone acetyltransferase (HAT), respectively [59]. Our data provide evidence that HMGA1 regulates gene expression in MDA-MB–231 cells through the modulation of both RSK2 and CBP activities. Therefore, we asked whether the two modifications were somehow interconnected. Since RSK2 activity relies on the activation of the ERK1/2 pathway, we evaluated whether this pathway was also involved in controlling H2B acetylation. Inhibition of the ERK1/2 pathway by the inhibitor U0126 also affected H2BK5ac in both MDA-MB–231 and MDA-MB–157 (Figure 7A). Moreover, RSKs inhibition by BI-1870 reduced H2BK5ac levels, and CBP/p300 inhibition by C646 downregulated H3S10ph levels (Figure 7B). Overall, these results suggest that in MDA-MB-231 cells, the presence of HMGA1 is relevant for both RSK2 and CBP activities and that together, these two factors cooperate in the expression of genes critical for tumor cell migration and aggressiveness. In order to clarify the interdependence of RSK2 and CBP we silenced in MDA-MB-231 cells both CBP and RSK2 evaluating the effects towards the gene expression of HMGA1, CBP, and RSK2 (Figure 7C). The silencing of both CBP and RSK2 did not affect HMGA1 gene expression level, and the silencing of RSK2 did not affect CBP gene expression. On the contrary, the CBP silencing caused a decrease in RSK2 gene expression level. In MDA-MB-157 cells the RSK/CBP interdependence is almost identical to that observed in MDA-MB-231. However, differently from what occurs in MDA-MB-231, the HMGA1 gene expression seems to be under the control of both RSK2 and CBP (Figure 7C).

### 2.6. HMGA1 Modulation of RSK2 and CBP Activities Is Likely to Occur by a Direct Mechanism and Not by Affecting Their Gene or Protein Expression Levels

We reasoned that HMGA1 could influence histone PTMs by different mechanisms: HMGA1 could control RSK2/CBP gene expression or modulate chromatin accessibility and recruitment onto DNA. To check whether HMGA1 could have a role in modulating the gene expression levels of these two enzymes, we silenced the expression of HMGA1 in MDA-MB–231 cells and looked at RSK2 and CBP mRNA levels. As shown in Figure 8A, upon HMGA1 downregulation, there was a slight downregulation of RSK2 mRNA expression, whereas CBP and p300 mRNAs were not downregulated but rather appeared slightly upregulated.

The downregulation of RSK2 mRNA opened up the possibility that the decrease in histone H3 S10 phosphorylation could be due to a decrease in the protein expression level of RSK2. The mechanism of RSK2 activation is complex; it relies on ERK activity, but the last kinase responsible for RSK2 activation is 3-phosphoinositide-dependent protein kinase-1 (PDK1), which phosphorylates RSK2 S227 in the N–terminal kinase domain [60]. Figure 8B shows that both the protein expression level of RSK2 and its phosphorylation at S227 did not significantly change after HMGA1 silencing. As additional proof that HMGA1 does not affect RSK2 activation, we checked the nuclear/cytoplasmic localization of RSK2. HMGA1 silencing did not affect the intracellular localization of RSK2, which, contrary to what we expected, turned out to be slightly, but not significantly, more nuclear (Figure 8C). 

These data strongly suggest that HMGA1 could affect both H3S10ph (and H3S28ph) and H2BK5ac by a mechanism involving the recruitment of RSK2/CBP onto DNA. It was previously demonstrated that HMGA1 is a key modulator of CBP recruitment onto chromatin [17]; therefore, we focused on the HMGA1/RSK2 interaction. To this end, we performed co-immunoprecipitation analyses (co–IPs) in the presence or absence of DNase I to discriminate between a direct and a chromatin–mediated interaction. As reported in Figure 8D, despite the presence of an unspecific band at a higher molecular weight with respect to RSK2 in the control experiment (lane 2), RSK2 was efficiently immunoprecipitated in the presence of DNA (lane 4), whereas it was not when DNA was digested (lane 5). Therefore DNA mediates the HMGA1/RSK2 interaction, but the two proteins lay very close to each other since the average length of the fragmented DNA (600 bp) encompasses approximately three nucleosomes (right part of panel 8D).

## 3. Discussion

The analysis of histone marks distribution among different BC molecular subtypes provides a specific chromatin signature defined by the pattern of activating or repressing PTMs. The chromatin distribution of these PTMs is indicative of the activation of distinct biological pathways and could also provide insights into clinical parameters (relapse-free survival and outcome) when applied to BC patients [61].

Since HMGA proteins bind nucleosome core particles [24] and HMGA1 expression inversely correlates with the overall survival in BC [62], our study aimed to evaluate possible epigenetic mechanisms by which this oncoprotein could contribute to the progression of TNBC, a particularly aggressive and heterogeneous intrinsic molecular subtype of BC with poor prognosis.

It has been previously demonstrated that: (i) HMGA proteins preferentially contact histones H3, H2A, and H2B, (ii) up to four HMGA1 molecules can bind to a single nucleosome, (iii) the binding occurs in a non-cooperative manner, probably on the “front face” of the nucleosome at the entry/exit site of the wrapped DNA, (iv) the acidic C-terminal tail of HMGA1 is not involved in this interaction, and (v) DNA/HMGA1 interactions are involved [24]. It was subsequently demonstrated that HMGA1 protein can bind to nucleosomal DNA and that it alters the nucleosome structure by changing the periodicity of nucleosomal DNA [63]. Moreover, HMGA proteins have already been linked to epigenetic alterations. In detail, HMGA2 has been linked to histone acetyl transferase (HAT) expression regulation in pancreatic ductal adenocarcinoma, which is in turn linked to increased levels of H3K9ac and H3K27ac in the fibrotic region. Downregulation of HMGA2 expression caused a reduction of acetylated H3 and sensitized cells to gemcitabine [64]. In addition, HMGA2 was shown to participate in the repression of the cadherin-1 (Cdh1) gene both by binding to the Cdh1 promoter and favouring the recruitment of the DNA (cytosine-5)-methyltransferase 3A (DNMT3A) and by binding to transcriptional repressor CTCF (CTCF) and displacing it from DNA, thus contributing to the EMT also via an epigenetic mechanism [65]. More recently, in a proteomic screening aimed at evaluating the interaction landscape of histones, it was demonstrated that HMGA1 was crosslinked with more than one core histone [66], thus suggesting intimate contact of HMGA1 with nucleosomes.

Our main findings in this work are the following: (a)HMGA1 silencing causes the downregulation of a series of histone post-translational modifications, i.e., mainly histone H3 S10 and S28 phosphorylation and histone H2B K5, K16, and K20 acetylation;(b)In the MDA-MB-231 cellular model these modification are due to the activity of RSK2 and CBP/p300;(c)The silencing of RSK2 and CBP phenocopies the effect of HMGA1 silencing towards a set of HMGA1 regulated genes while p300 silencing does not;(d)Impairment of RSK2 activity by means of the RSKs inhibitor BI-D1870 is responsible for a decrease of histone H3 S10 phosphorylation at specific regulatory regions of HMGA1 regulated genes;(e)HMGA1 does not modulate the gene/protein expression level of RSK2 and CBP and HMGA1 does not modulate the nuclear localization of RSK2 neither its activation status (i.e., RSK2 S227 phosphorylation level);(f)Impairment of RSK2 activity by means of the RSKs inhibitor BI-D1870 is responsible for MET;(g)HMGA1 and RSK2 interact in a DNA-dependent way.

Altogether these data suggest a role of HMGA1 in modulating the activity of RSK2 and CBP at chromatin level. H3S10ph is a convergence point of several signalling cascades (reviewed in [67]). H3S10ph and H3S28ph affect the adjacent docking sites, H3K9me3 and H3K27me3, for HP1 and Polycomb group proteins [38,40] and are usually referred to as epigenetic molecular switches. It has also been demonstrated that H3S10ph cooperates synergistically with histone acetylation in providing a transcriptionally competent environment [39].

RSK2 has emerged as a specific therapeutic target for metastatic TNBCs, and the use of an RSK2 inhibitor has shown to reduce cell proliferation, survival in a non-adherent environment, and migration [68]. Inhibition of RSK2 suppresses tumor-initiating cell growth and promotes cell death [69]. Some authors showed that 85% of TNBC patients exhibit activation of RSK2 (S227ph) in a panel of high-grade BC samples. Moreover, RSK2 inhibitor treatment could overcome the side effects of MEK/ERK inhibitors in clinical trials: in fact, the efficiency of MEK/ERK inhibition is often lowered by feedback loop activation of the phosphoinositide-3-kinase–protein kinase B/AKT (PI3K-PKB/AKT) pathway [70].

CBP/p300 are versatile HATs with context-dependent functions, and indeed, the large multiprotein complexes they belong to usually dictate their functional outcome. In fact, CBP/p300 mediates the tumor suppressor function of p53 [71], Forkhead box protein O3 (FOXO3A) [72] and Breast cancer type 1 susceptibility protein (BRCA1) [73], while on the other hand, together with c-Myc [74], c-Myb [75] and androgen receptor (AR) co-activators [76], they can promote cancer progression. HAT inhibition has been pursued as a therapeutic tool in the fight against cancer: bisubstrate inhibitors (mimicking the two HAT substrates, the acetyl-CoA and the lysine-containing peptide connected by a linker), natural compounds (anacardic acid, curcumin, garcinol), small molecule inhibitors (C646), and protein/protein interaction inhibitors are currently being investigated in different tumor types, despite such limitations as instability, low cell permeability, selectivity, and low potency [77]. In vivo, the CBP/p300 inhibitor L002 suppresses tumor growth and recurrence in MDA-MB-468 mouse xenograft models [78]. Although both CBP and p300 can modulate H2BKac, we found CBP to be responsible for HMGA1 signature regulation, and these results underline the different specificities gained by different HATs.

Our results confirm the cooperation between RSK2 and CBP in histone modification regulation in the TNBC cellular model. In detail, levels of H3S10ph and H2BK5ac are reciprocally modulated by HAT and kinase activities and both decreased upon HMGA1 silencing while the gene/protein expression level of RSK2 and CBP seems not be affected. Moreover, inhibition of CBP or RSK2 mirrors the effects obtained by HMGA1 silencing, i.e., the regulation of a common set of genes and gross morphological cellular changes. Interestingly, we found that the silencing of RSK2 does not modulate CBP expression, but RSK2 activity impairment leads to the down regulation of histone H2B K5 acetylation. In our opinion these experimental evidences suggest that the RSK2/CBP interplay happens directly onto chromatin and is not due to an effect at transcriptional level, effect that cannot be excluded when CBP is silenced since we demonstrated that upon CBP silencing RSK2 gene expression is lowered.

These data should prompt us to rethink the role of HMGA proteins as chromatin architectural factors: they should be regarded not only as DNA–binding factors that directly bind to DNA and modify its structure, thus favouring the landing of other transcription factors, but also as multifaceted chromatin organizers (Figure 9). We demonstrated that the silencing of HMGA1 causes a reduction of histone H3S10ph and histone H2BK5ac and proved that these two modifications are dependent on the activity of RSK2 and CBP/p300. It was previously demonstrated that both ERK1/2 and RSK2 were promoter-bound kinases and suggested that the positioning of kinases in such locations could allow them to target substrates not otherwise accessible [79]. CBP/p300 is a well known transcriptional coactivator, it is recruited onto chromatin throughout interaction with transcription factors and is generally involved in the organization of a bridge between the promoter/enhancer bound factors and the general transcription factors (reviewed in [80]). Several mechanisms could explain this HMGA1-mediated epigenetic effect. The simplest one is that given the ability of HMGA1 to bind to nucleosomes it could organizes the assembly of a nucleosome-bound complex that in turn could recruit RSK2 and CBP onto chromatin. However, HMGA1 has been also demonstrated to be able to enhance the binding of transcription factors throughout its ability to confer conformational changes to transcription factor consensus sequences or via direct protein-protein contacts, for example facilitating nuclear factor-kappa-B (NF-κB) DNA binding [81] or binding to serum response factor (SRF) and changing its DNA binding properties [82]. Moreover, it was also demonstrated that HMGA1 is involved in perturbing DNA topology and influencing long-range enhancer transcriptional activity by organizing DNA loops [83]. Our data demonstrated that the final output of HMGA1 action is a change in the post-translational modifications of histones and this implies that the effect of HMGA1 is towards chromatin but the exact mechanism behind this event is not still clarified and we do not exclude the possibility that multiple HMGA1-mediated mechanisms act at the same time.

The possibility of a chromatin-mediated mode of action of HMGA1 opens the need to rethink the possible effect of HMGA1 PTMs towards chromatin binding. HMGA proteins are modulated by a large amount of PTMs (reviewed in [84]), some of which affect the entire bulk of HMGA proteins [85,86]. Most of the work carried out to decipher the role of HMGA PTMs has been performed by evaluating their effect on direct DNA binding [84]. Intriguingly, in vivo data regarding the effect of HMGA1 phosphorylation have demonstrated exactly the opposite of what has been demonstrated in vitro: HMGA1 phosphorylation increases the chromatin residence time of HMGA1 [87], data in sharp contrast to the well–documented decrease in DNA–binding affinity upon HMGA phosphorylation [84].

HMGA1 is usually expressed at very high level in cancer cells (up to a ratio of 1/5 with histone H1 [88]. It displaces histone H1, binds to DNA and chromatin (nucleosomes) and in this way it facilitates the assembly of chromatin/DNA-bound macromolecular complexes, which are mainly, but not only, involved in transcriptional activation. This chromatin architectural activity could allow HMGA1 to participate in the delivery toward chromatin of signals from other oncogenic pathways. Limiting our discussion to histone H3 S10 phosphorylation, it is worthwhile to mention that RSK2 is not the only kinase able to deposit this PTM. MSK1 is known to be a downstream effector of the RAS pathway and to be recruited onto chromatin as part of macromolecular complexes that, for instance include CBP/p300, NF-κB, and transcription factor AP-1 (AP1), or ERK and progesterone receptor (PR) [89,90]. Moreover, MSK1 was shown to be activated also in a protein kinase C (PKC) dependent way [91]. Inhibitor of nuclear factor kappa-B kinase subunit alpha (IKK-alpha) is recruited onto chromatin by means of a NF-κB mediated mechanism and therein is involved in the phosphorylation of histone H3 S10 [92,93]. IKK-alpha is known to be activated by a plethora of different stimuli (reviewed in [94]). Aurora kinase B, is mainly involved in mitosis, but it has a role also in interphasic cells where, for instance it is recruited to several promoters by a thyroid hormone (T3)-dependent mechanism [95]. Moreover, as concerns AURKB, it is worthwhile to notice that HMGA1 is involved in the gene transcriptional regulation of AURKB. In addition, RSK2 has a prominent role in the activation of the transcription factor Y-box binding protein-1 (YB-1) which is a main factor leading to the development of basal-like cancer [69,96,97]. Albeit in a different cellular model, i.e., in non-small-cell lung cancer (NSCLC) cells A549, it is intriguing that both HMGA1 and YB-1 have been demonstrated to bind to the promoter of cyclin D1 and to be responsible for its transcriptional regulation [98,99]. In breast cancer cells it was demonstrated that chromatin remodelling is upstream of YB-1 landing onto promoters [97]. Albeit at a speculative level, HMGA1 could be one of the chromatin factors cooperating with YB-1 in breast cancer onset and development. We demonstrated that HMGA1 is involved in maintaining a mesenchymal phenotype and to confer aggressive and stemness traits to TNBC cells [29] and that part of this activity is performed by modulating a downstream mediator of the Hippo pathway, i.e., YAP [54]. In addition, we showed that a set of HMGA1-dependent genes constitute a molecular signature with prognostic value in breast cancers [28] and HMGA1 expressing cells secreted glycosylated factors involved in modulating cell motility and invasiveness [34]. Very recently we demonstrated that HMGA1 is involved in modulating breast cancer cells nuclear stiffness trough a mechanism involving histone H1 [35] and that it drives also angiogenic properties acting in cooperation with forkhead box protein M1 (FOXM1) [100]. Overall, HMGA1 exerts pleiotropic oncogenic effects in MDA-MB-231 cells, which have been extended to other TNBC cells. In this work we continued this characterization of the oncogenic role of HMGA1 exploring an additional mechanism, i.e., the contribution of HMGA1 to the epigenetic status of cancer cells. We focused on MDA-MB-231 cells because they are triple-negative and basal-like breast cancer cells and we previously showed, by performing a bioinformatic analysis of a primary breast cancer public microarray data collection (1881 different samples), that HMGA1 mRNA levels were higher in the basal-like than the HER2+, the luminal A and B, and normal-like subtypes [29]. For sure one of the limitation of this study is that it mainly relies upon a single cellular model, however, we speculate that it would not be surprising to find out that HMGA1 concurs with the same mechanism in conveying to chromatin signals originating from other oncogenic pathways that exploit different actors.

## 4. Materials and Methods 

### 4.1. Cell Cultures and Treatments

The breast cancer cell lines used in this study are: adenocarcinoma MDA-MB-231 (TNBC, mesenchymal like/claudin low-KRAS mutation); medullary carcinoma MDA-MB-157 (TNBC, mesenchymal like/claudin low); adenocarcinoma MDA-MB-468 (TNBC, basal like—amplification of EGFR). Cell lines were kindly provided by the laboratory of Prof. G. Del Sal (Laboratorio Nazionale CIB (LNCIB) Area Science Park, Trieste, Italy). Cells were cultured as previously described [101]: MDA-MB-231 and MDA-MB-157 cells were grown in Dulbecco’s Modified Eagle Medium (DMEM) plus 10% tetracycline-free fetal bovine serum (FBS), MDA-MB-468 cells in Roswell Park Memorial Institute (RPMI) 1640 medium plus 10% tetracycline-free FBS. Silencing experiments were carried out essentially as previously described [28]; briefly, cell lines were transfected with 100 nM siRNAs with Lipofectamine^TM^ RNAiMAX reagent (Invitrogen, Carlsbad, CA, USA) and collected after 72 h. All siRNA sequences are reported in Appendix A. All the inhibitors used are listed in Appendix A.

### 4.2. qRT-PCR, Cell proliferation (MTS) assay, Wound Healing Assay, Transwell Migration Assay, SDS-PAGE, and Western Blot Analyses

These analyses were performed with conventional procedures essentially as previously described [28,29]. Briefly, total RNA was extracted using Trizol reagent (Invitrogen, Carlsbad, CA, USA), treated with DNase I (Invitrogen, Carlsbad, CA, USA), and subsequently purified with a conventional phenol-chloroform extraction method. qPCR was performed using iQTM SYBR Green Supermix (Bio-Rad Laboratories, Hercules, CA, USA), the data obtained were analysed with the Bio-Rad CFX Manager software and the relative gene expression was calculated by the ΔΔCt method. The primer sequences are reported in Appendix A. Cell proliferation was monitored using the MTS assay. 7000 cells were seeded into 96-well plate and left grown for 72 h. Cell viability was determined by CellTiter 96 Aqueous One Solution Assay (Promega, Madison, WI, USA) according to the manufacturer’s instructions. Briefly, after 0, 24, 48, or 72 h of incubation in the growth medium with or without BI-D1870 10 μM, 120 μL of MTS solution (MTS reagent diluted 1:6 in PBS containing 4.5 g/L glucose) was added into each well, the plate was incubated for 2 h at 37 °C in 5% CO_2_ atmosphere, and then absorbance at 480 nm was recorded with an INFINITE M200 PRO microplate reader (Tecan Group Ltd., Männedorf, Switzerland). Wound healing assays were performed on 3 mL cell culture plates with cells at about 80% confluence. The cells were scraped with a 200 μL tip, and wound closure was followed for 8 h. Measurements were made calculating the area in the middle part of the wounds selecting as much as possible straight and homogeneous zones. Reference points were used to select starting and ending lines for the area measurements. For transwell migration assays, 24-well polyethylene terephthalate (PET) inserts were used (8.0 μm pore size, Falcon (Corning), Corning, NY, USA) with matrigel-coated filters. 40,000 cells were seeded and after 20 h migrated cells were fixed in PFA 4% and stained with Crystal Violet 0.5% (Sigma Aldrich, St. Louis, MO, USA). For sodium dodecyl sulphate-polyacrylamide gel electrophoresis (SDS-PAGE) analyses, cells were washed in chilled phosphate-buffered saline (PBS) buffer and lysed using SDS sample buffer. SDS-PAGE analyses were made with T = 15% SDS Tris-Tricine polyacrylamide minigels. For western blot analyses, protein where transferred onto nitrocellulose membranes, stained with red ponceau, images acquired with a densitometric scanner, and then subjected to antibodies recognition. The antibodies used are listed in Appendix A.

### 4.3. Immunofluorescence (IF) Analyses

Cells cultured on glass coverslips were rinsed twice in PBS and fixed for 20 min at room temperature (RT) with 4% paraformaldehyde (PFA) in PBS pH 7.2. Cells were washed three times in PBS and then 0.1 M glycine was added for 5 min at RT. After three washes in PBS, cells were permeabilised with 0.3% Triton X for 5 min at RT. Cells were washed three times and unspecific sites were blocked with 0.5% (*w*/*v*) bovine serum albumin (BSA) in PBS for 30 min at RT. Each coverslip was incubated for 90 min in a wet chamber with the primary antibodies diluted in blocking solution. After three washes with PBS, samples were incubated with secondary antibody diluted in blocking solution for 60 min in a wet chamber. Coverslips were washed three times in PBS and incubated 5 min with 0.2 μg/mL Hoechst (Sigma Aldrich, St. Louis, MO, USA) in PBS. Images were acquired with an Eclipse Ti inverted research microscope (Nikon, Tokyo, Japan) equipped with a NIS-Elements software. The antibodies used for immunofluorescence analyses are listed in Appendix A. The nuclear/cytoplasmic mask was obtained using Hoechst staining to identify the nucleus and subtracting this region from the whole cellular area stained with the RSK2 antibody to identify the cytoplasmic portion of the cell.

### 4.4. Co-immunoprecipitation Assay 

Co-immunoprecipitations were carried out essentially as previously described [102]. A 150-mm plate of cells at sub-confluence, was rinsed twice in cold PBS and collected with scraper. Cells were pelleted (4 °C, 450× *g*, 5 min) and washed again in PBS. Supernatant was discarded and cells resuspended in lysis buffer (25 mM Tris/HCl pH 8, 0.5% *v*/*v* NP40, 125 mM NaCl, 10% *v*/*v* glycerol) supplemented with 1/1000 (*v*/*v*) Phenylmetylsulfonyl fluoride (PMSF, saturated solution in isopropanol), 1 mM NaVO3, 5 mM NaF, 10 mM sodium butyrate and 1/100 (*v*/*v*) protease inhibitor cocktail (Sigma Aldrich, St. Louis, MO, USA) and incubated 15 min on ice. Samples were then sonicated at 30% of potency for 60 s (10 s ON, 30 s OFF) with a Branson Digital Sonifier 250 (Branson Ultrasonics, Danbury, CT, USA), centrifuged (4 °C, 8385× *g*, 10 min), and supernatant collected and stored at −80 °C. 30 μL of Protein G Sepharose (GE Healthcare, Chicago, IL, USA) were used for each co-immunoprecipitation. Beads were washed three times with Tris/HCl 50 mM pH 7 and incubated for one hour at 4 °C with 4 μg of α–HMGA1 antibody or IgGs (#ab37415, Abcam, Cambridge, UK) used as negative control, while the volume was adjusted with Tris/HCl 50 mM pH 7 to 500 μL. After centrifugation (4 °C, 1027× *g*, 1 min) supernatant was discarded and BSA 1 mg/mL in lysis buffer was added at 4 °C for 1 h. Beads were then washed three times in lysis buffer (4 °C, 1027× *g*, 1 min). 500 μg of lysates, quantified by the bicinchoninic acid (BCA) method, were adjusted to 1 mL with lysis buffer and incubated with beads for 3 h. To verify that the interaction between proteins was not mediated by DNA a parallel experiment was performed using DNase I. For each co–IP, 500 μg of lysate adjusted to 500 μL with lysate buffer and DNase digestion buffer 10x (Invitrogen, Carlsbad, CA, USA) were digested with DNase I (1000 U—Sigma Aldrich, St. Louis, MO, USA) for 20 min at 37 °C. An aliquot of 50 μL of lysate either digested or not (control without DNase) was kept and made 1% sarcosile and 25 mM ethylenediamine tetraacetic acid (EDTA). The aliquot was digested with 10 μg of RNase A (Sigma Aldrich, St. Louis, MO, USA) for 1 h at 55 °C and later with 23 μg of Proteinase K (Sigma Aldrich, St. Louis, MO, USA) for 1 h at 55 °C. 10 μL were conditioned with an equal volume of Blue/Orange DNA loading dye (Promega, Madison, WI, USA) and loaded to 1.5 % (*w*/*v*) agarose in tris-acetate-EDTA (TAE) buffer (1/25000 GelRed Biotium, Fremont, CA USA). Samples were subjected to electrophoretic separation for 40 min at 50 V. The visualization was done with a Gel Doc instrument (Bio-Rad Laboratories, Hercules, CA, USA). After the incubation with lysates, beads were centrifuged (4 °C, 1027× *g*, 1 min), supernatants were discarded and beads washed three times in lysis buffer. 20 μL of SDS sample buffer were added to each sample, heated for 5 min at 96 °C and analysed by WB.

### 4.5. Chromatin Immunoprecipitation

Cells were fixed with 1% formaldehyde in serum–free medium, washed twice with PBS at room temperature (RT) and incubated with 125 mM Glycine/PBS (RT) for 5 min. After two washes with cold PBS, cells were gently scraped with cold PBS and pelleted (4 °C, 450× *g*, 10 min). Pellet was resuspended in 1 mL of cold Lysis Buffer (5 mM PIPES pH 6.8, 85 mM KCl, 0,5% NP40 (*v*/*v*)) supplemented with a protease inhibitor cocktail (PIC—Sigma Aldrich, St. Louis, MO, USA) and incubated for 10 min at 4 °C. Afterwards, nuclei were centrifuged (4 °C, 2100× *g*, 10 min) and resuspended in 1 ml cold RIPA 100 buffer (20 mM Tris-HCl pH 7.5, 100 mM NaCl, 1mM EDTA, 0,5% NP40 (*v*/*v*), 0,05% Na-deoxycholate (*v*/*v*), 0,1% SDS (*w*/*v*)) supplemented with PIC. Samples were sonicated (Digital Sonifier 250, Branson Ultrasonics, Danbury, CT, USA) at 10% potency for 1 min (10 s ON, 30 s OFF on ice, for six times). DNA fragments were checked by electrophoretic analyses. DNA samples were quantified with Qubit fluorometer (Invitrogen, Carlsbad, CA, USA—Quant-iTTM dsDNA BR Assay kit, #Q32850). For each ChIP, 80 µL of Protein A/G 1:1 mixture (Protein G SepharoseTM 4 Fast Flow, #17-0618-01; Protein A SepharoseTM 4 Fast Flow #17-0974-01, GE Healthcare, Chicago, IL, USA) were washed (4 °C, 1000× *g*, 2 min) twice with cold RIPA 100 and blocked by with 1 mL RIPA 100 (1 mg/ml Salmon Sperm DNA, 1 mg/mL BSA) for 1 h at 4 °C. Equal amounts of cell lysate (2 Petri dishes) for each sample were precleared with blocked A/G beads for 1 h at 4 °C. After incubation, samples were centrifuged (4 °C, 1000× *g*, 2 min) and the precleared lysate was incubated o/n at 4 °C with 3 µg of specific antibody or IgGs (#ab37415, Abcam, Cambridge, UK) as negative control. An aliquot from precleared lysate was conserved as input. Immunoprecipitation (IP): for each IP, 40 µL of A/G beads blocked o/n as described above, were washed twice with RIPA 100 and incubated with the lysate/antibody solution for 3 h at 4 °C. Beads were washed twice with cold RIPA 100 supplemented with PIC, twice with cold RIPA 250 (20 mM Tris-HCl pH7.5, 250 mM NaCl, 1mM EDTA, 0,5% NP40 (*v*/*v*), 0,05% Na-deoxycholate (*v*/*v*), 0,1% SDS (*w*/*v*)) supplemented with PIC, and once with cold LiCl solution (10 mM Tris/HCl pH8, 1 mM EDTA, 250 mM LiCl, 0,5% Na-deoxycholate (*v*/*v*), 0,5% NP40 (*v*/*v*)) for 10 min at 4 °C. Afterwards, beads were washed with Tris/EDTA buffer (TE) (10 mM Tris/HCl pH 8, 1 mM EDTA pH 8) and resuspended in 100 mL TE. IP and input samples were complemented with 400 µL water and 20 µL NaCl 5 M. All samples were incubated at 65 °C o/n on a thermomixer. Samples were treated with RNase A and Proteinase K, phenol/chloroform extracted, and analysed by qPCR. 

### 4.6. Statistical Analysis

Experimental data were evaluated by a two-tailed Student’s *t*-test, which provided means and standard deviations. WB histograms represent densitometry values of normalized bands versus densitometry of total lysates of ponceau-stained membrane. For both WB and qRT-PCR, the values were normalized to the control samples. Immunofluorescence (IF) data were analysed by the Shapiro-Wilk test to check their probability distribution: with a *p*-value less than 0.05, they were considered not normally distributed, and non-parametric statistics were used, such as the Mann-Whitney test. Data are presented as boxplots, with median and whiskers from the 5th to 95th percentiles.

## 5. Conclusions

In conclusion, we provide a collection of evidence that the HMGA1 protein could promote the aggressiveness of BC cells by an epigenetic mechanism. More specifically, our data allow us to speculate that HMGA1 could be involved in the recruitment of histone modifiers onto chromatin. Given the difficulty of specifically targeting HMGA1 in breast cancer, we suggest that HMGA1 expression could steer the choice of therapeutic approaches towards epigenetic–targeting options. Further work has to be done to in–deep evaluate the role of HMGA1 in influencing the epigenetic status of cancer cells.

## Figures and Tables

**Figure 1 cancers-11-01105-f001:**
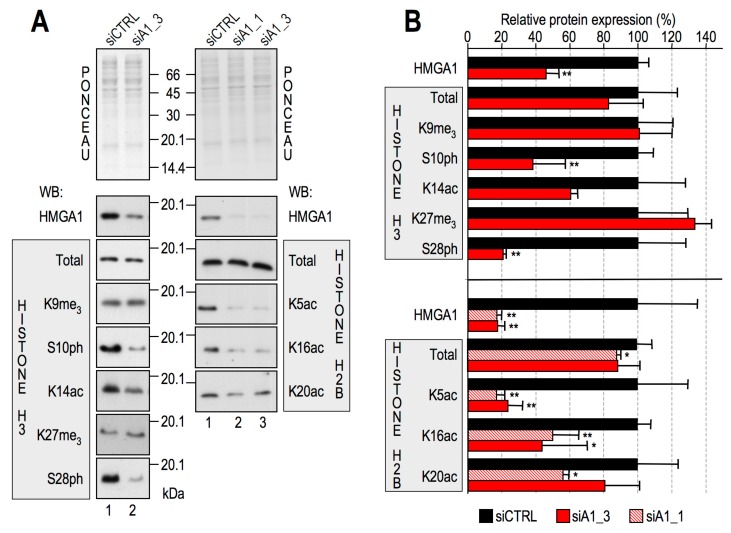
High Mobility Group A1 (HMGA1) affects H3 and H2B post-translational modifications in MDA-MB-231 cells. (**A**) Western blot (WB) analyses of MDA-MB-231 cells transfected with control (siCTRL—lanes 1) and HMGA1–targeting (siA1_3 or siA1_1—lanes 2 and 3) small interfering RNAs. Representative Red Ponceau-stained membranes for loading and quantification control are shown. Molecular weight markers (kDa) are indicated. (**B**) Densitometric analyses of WB signals. Means and standard deviations are reported (*n* = 3). Statistical significances (*t*-test) are indicated: (*: *p* < 0.05; **: *p* < 0.01).

**Figure 2 cancers-11-01105-f002:**
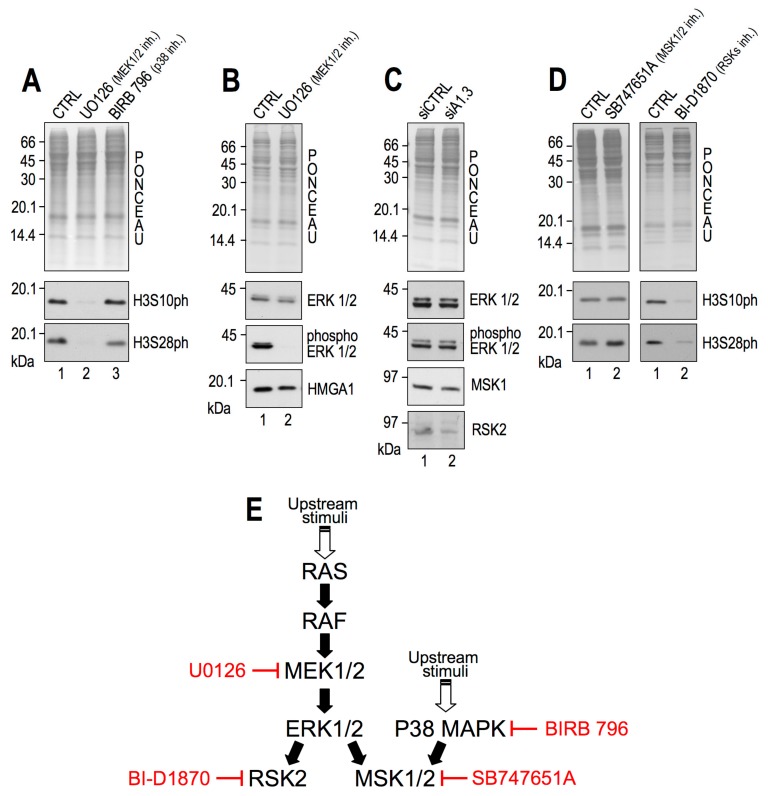
H3S10ph and H3S28ph are mediated by the extracellular-signal-regulated kinase/ribosomal protein S6 kinase alpha-3 (ERK/RSK2) pathway in MDA-MB-231 cells. Western blot (WB) analyses performed with MDA-MB–231 lysates. (**A**) Cells treated for 24 h with 10 µM U0126 (lane 2), 10 µM BIRB796 (lane 3), or dimethyl sulfoxide (DMSO) as control (CTRL, lane 1). (**B**) Cells treated for 24 h with 10 µM UO126 (lane 2) and DMSO as control (CTRL, lanes 1). (**C**) Cells transfected with control (siCTRL, lane 1) or HMGA1–targeting (siA1_3, lane 2) small interfering RNAs. The HMGA1 silencing efficacy of siA1_3 treatment is shown in Figure 1. (**D**) Left side: cells treated for 24 h with 10 µM SB747651A (lane 2) and DMSO as control (CTRL, lanes 1). Right side: cells treated for 24 h with 10 µM BI–D1870 (lane 2) and DMSO as control (CTRL, lanes 1). Representative red ponceau-stained membranes are shown as loading and quantification controls. Molecular weight markers (kDa) are indicated on the left. All WB experiments were performed in biological triplicate, providing consistent results. Representative images are shown. (**E**) Summary of all the inhibitors used and their targets within the studied pathways.

**Figure 3 cancers-11-01105-f003:**
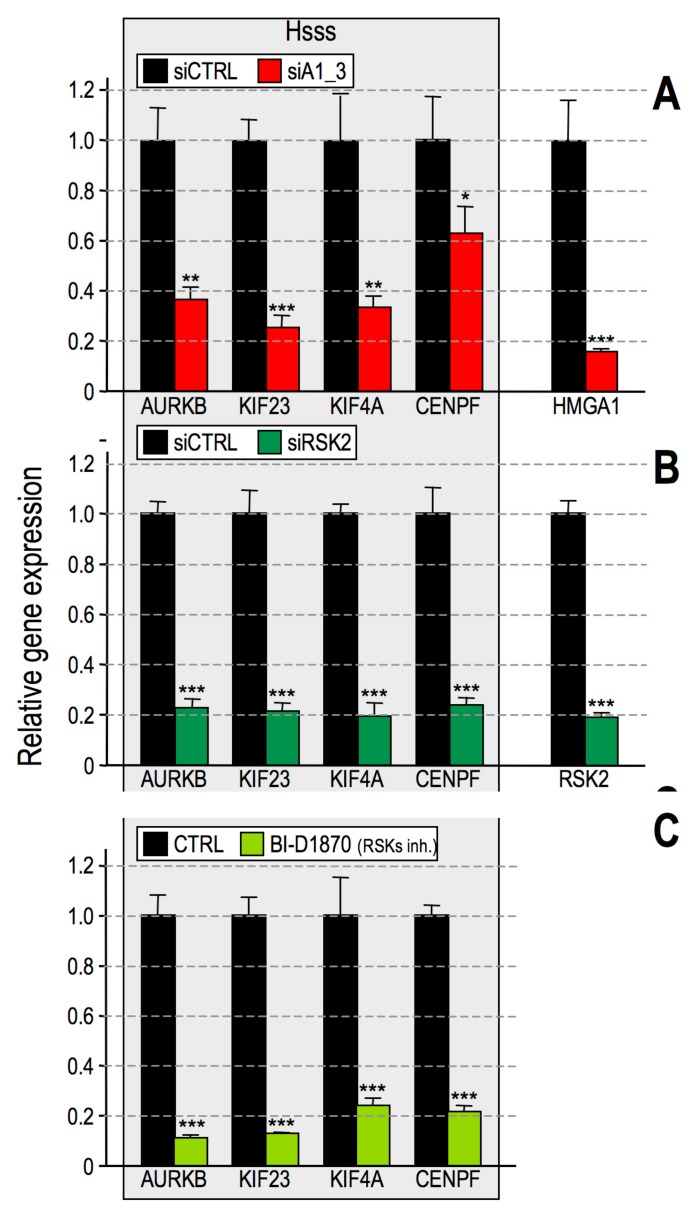
High Mobility Group A1 (HMGA1) and ribosomal protein S6 kinase alpha-3 (RSK2) regulate a common set of genes. Gene expression analyses performed by qRT-PCR on MDA-MB-231 cells transfected with (**A**) control (siCTRL) or HMGA1–targeting (siA1_3) small interfering RNAs, (**B**) control (siCTRL) or RSK2–targeting (siRSK2) siRNA, or (**C**) treated for 24 h with BI–D1870 10 μM or dimethyl sulfoxide (CTRL) as a control. siRNA-treated cells were harvested after 72 h. Expression of the HMGA1 subset signature (Hsss - AURKB, KIF23, KIF4A, CENPF), RSK2, and HMGA1 genes was analysed using glyceraldehydes-3-phosphate dehydrogenase (GAPDH) or cyclophilin-33 (CYP33) as internal reference genes. Data are represented as relative gene expression values with respect to control samples. Standard deviations (*n* = 3) and statistical significance (*t*-test) are indicated (*p*-values: *: *p* < 0.05; **: *p* < 0.01; ***: *p* < 0.001).

**Figure 4 cancers-11-01105-f004:**
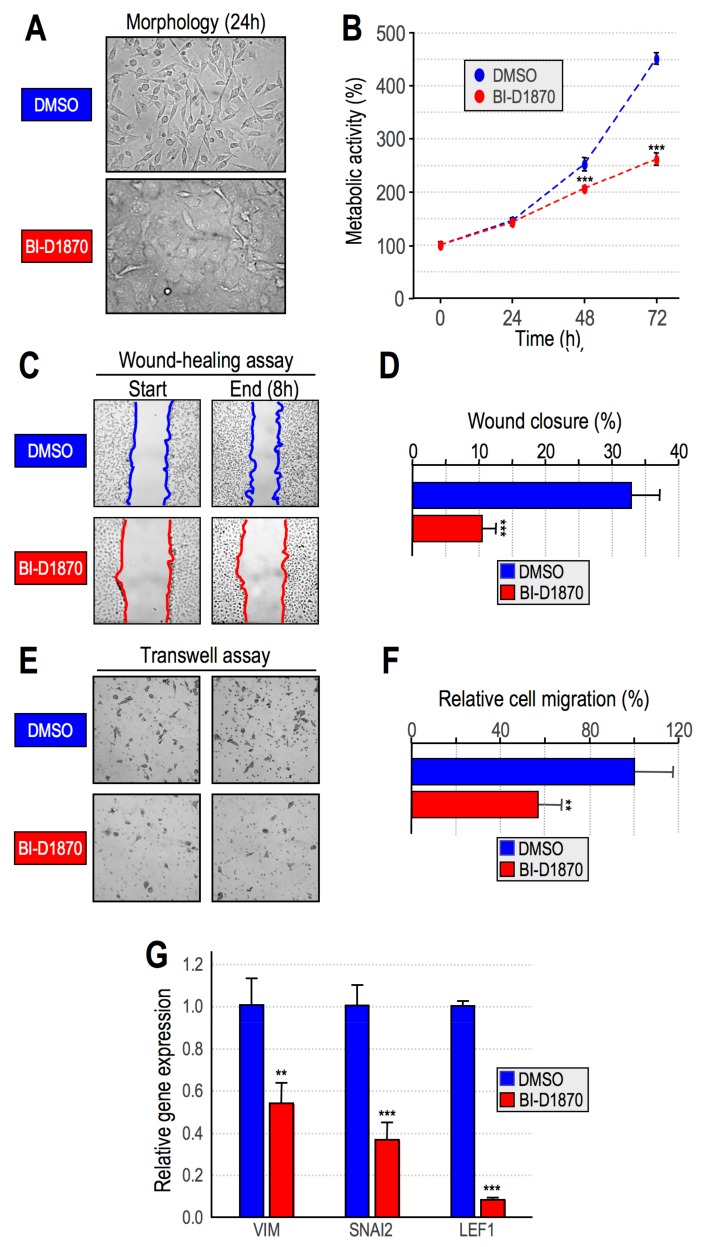
Ribosomal protein S6 kinase alpha-3 (RSK2) activity is essential in MDA-MB–231 cells to maintain a mesenchymal phenotype and cell motility. (**A**) Optical microscope images of MDA-MB-231 cells treated for 24 h with BI-D1870 10 µM. Dimethyl sulfoxide (DMSO)–treated cells are shown as a control. (**B**) [3-(4,5-dimethylthiazol-2-yl)-5-(3-carboxymethoxyphenyl)-2-(4-sulfophenyl)-2H-tetrazolium]-based (MTS) assay of MDA-MB-231 cells treated with BI-D1870 10 µM in comparison with DMSO treatment as a control (*n* = 6). (**C**) Wound–healing assay of MDA-MB-231 cells treated with BI-D1870 10 µM. DMSO–treated cells are shown as a control. Representative images of start (0 h) and end (8 h) time points are reported. (**D**) Quantitative evaluation of the wound–healing assay. Data are represented as the means of the percentage of wound closure relative to the start time point. The experiment was performed in technical duplicate and biological triplicate. (**E**) Transwell assay of MDA-MB-231 cells treated with BI-D1870 10 µM. DMSO–treated cells are shown as a control. Two representative images are reported for each treatment. (**F**) Quantitative evaluation of the Transwell assay (*n* = 4, technical sextuplicate). (**G**) Gene expression analyses performed by qRT-PCR on MDA-MB-231 cells treated for 24 h with BI–D1870 10 μM or DMSO as a control. Expression of the epithelial to mesenchymal transition (EMT) markers vimentin (VIM), zinc finger protein SNAI2 (SNAI2), and lymphoid enhancer-binding factor 1 (LEF1) genes was analysed using glyceraldehydes-3-phosphate dehydrogenase (GAPDH) as internal reference genes. Data are represented as relative gene expression values with respect to control samples (*n* = 3). Standard deviations and statistical significance (*t*-test) are indicated (*p*-values: **: *p* < 0.01; ***: *p* < 0.001).

**Figure 5 cancers-11-01105-f005:**
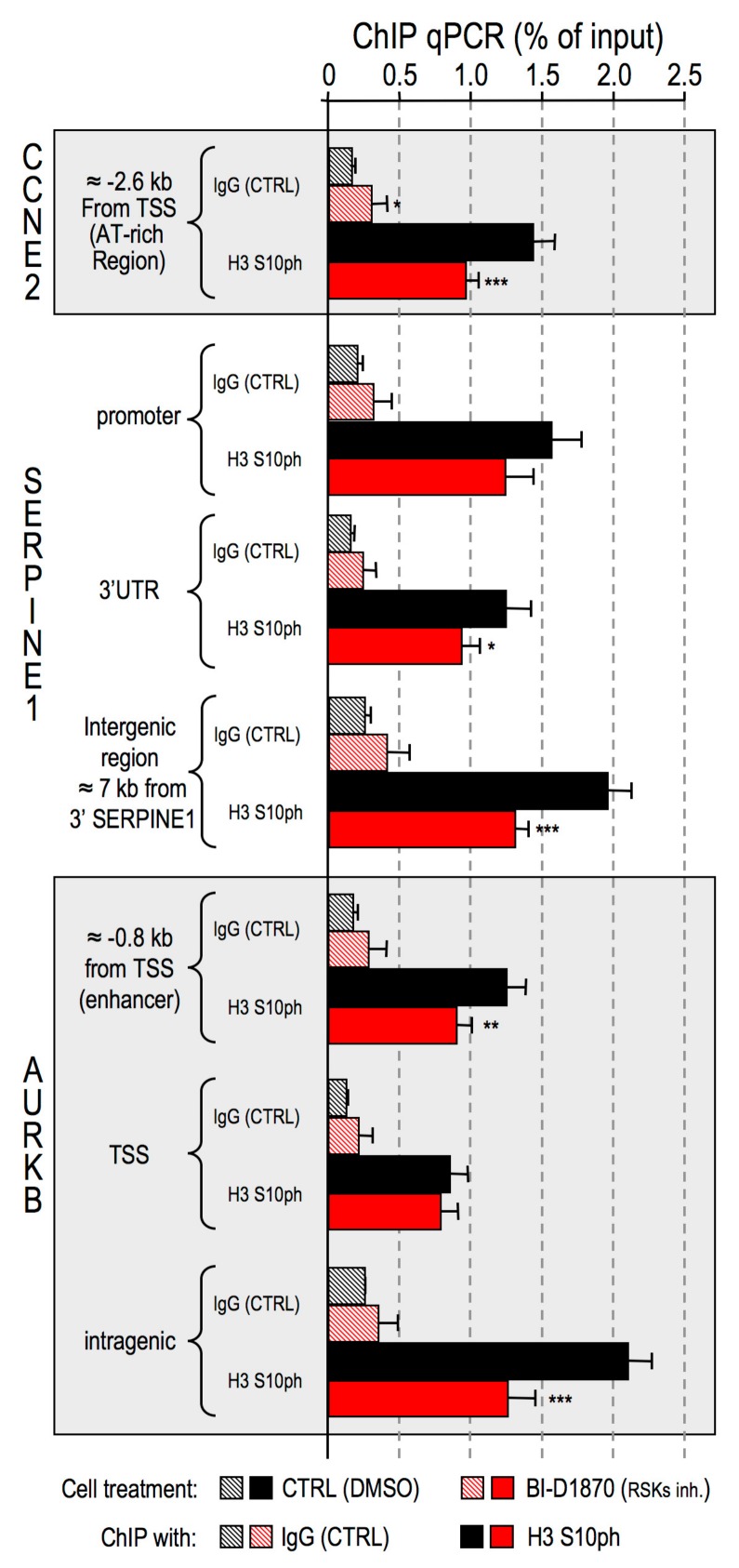
Ribosomal protein S6 kinase alpha-3 (RSK2) is responsible for H3S10 phosphorylation at the level of genes that are transcriptionally modulated by High Mobility Group A1 (HMGA1). Chromatin immunoprecipitation (ChIP)-qPCR experiments performed on MDA-MB-231 cells treated with the RSK2 inhibitor BI-D1870. Cell lysates from control and BI–D1870–treated cells were immunoprecipitated both with α–histone H3S10ph antibody and non-specific immunoglobulins (IgG) as a control (CTRL). Immunoprecipitated DNA was amplified with primers targeting the indicated genomic region within three HMGA1 target genes (G1/S-specific cyclin E2, CCNE2; plasminogen activator inhibitor 1, SERPINE1; aurora kinase B, AURKB). Enrichments for the individual antibodies used are plotted as percentage of input DNA. Means, standard deviations (*n* = 3), and statistical significance (*t*-test) are indicated (*: *p* < 0.05; **: *p* < 0.01; ***: *p* < 0.001).

**Figure 6 cancers-11-01105-f006:**
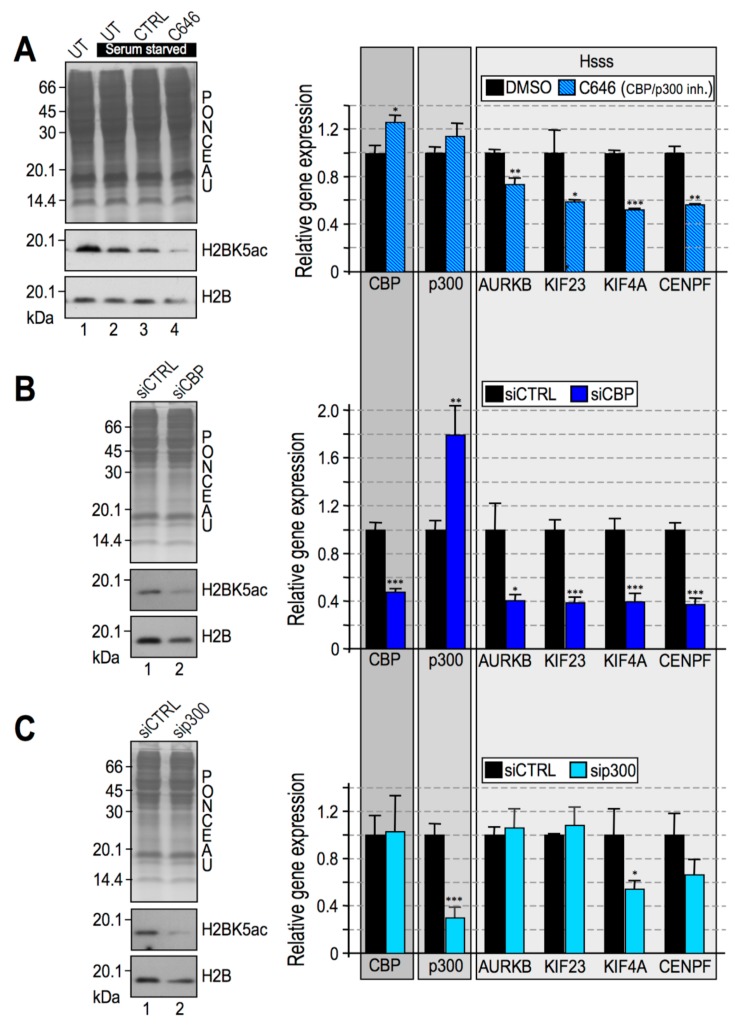
CREB-binding protein (CBP) is linked to the gene expression regulatory role of High Mobility Group A1. (**A**, **B**, and **C**—left side) Western blot (WB) analyses performed with MDA-MB–231 cell lysates. Cells were grown with complete medium (lane 1) or serum-starved medium (lanes 2, 3 and 4) for 24 h before treatment with 20 μM C646 or 0.2% dimethyl sulfoxide (DMSO) as a control (CTRL). UT (untreated) represents cells treated neither with C646 nor with DMSO (**A**—left side). MDA-MB–231 cell lysates silenced for CBP and p300 (siCBP and sip300) and their relative controls (siCTRL) (**B** and **C**—left side, respectively). All WB experiments were performed in biological triplicate, providing consistent results. Representative images are shown. (**A**, **B**, and **C**, right side) qRT-PCR analyses of serum-starved MDA-MB-231 cells treated for 24 h with 20 μM C646 (**A**, right side) MDA-MB-231 cells silenced for CBP (**B**, right side) or p300 (**C**, right side) and their respective controls (DMSO and siCTRL). The four HMGA–signature genes (Hsss) were analysed together with CBP and p300. Glyceraldehydes-3-phosphate dehydrogenase (GAPDH) or cyclophilin-33 (CYP33) were used as internal reference genes. Mean, standard deviations (*n* = 3), and statistical significance (*t*-test) are indicated (*: *p* < 0.05; **: *p* < 0.01; ***: *p* < 0.001).

**Figure 7 cancers-11-01105-f007:**
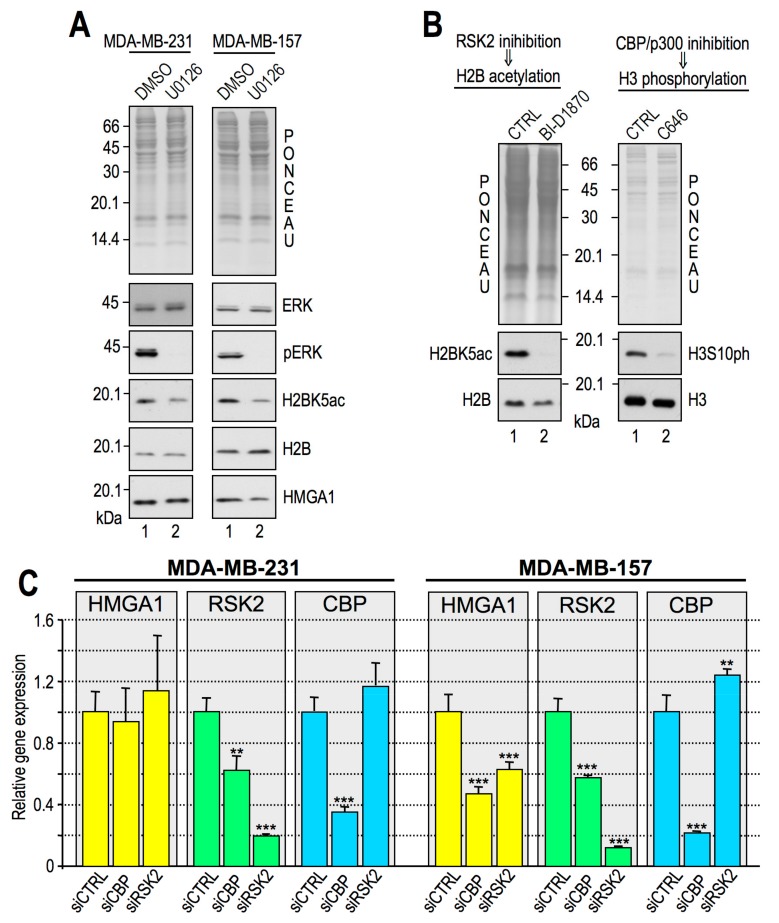
Ribosomal protein S6 kinase alpha-3 (RSK2) and CREB-binding protein (CBP) are interconnected epigenetic writers in MDA-MB–231 cells. Western blot (WB) analyses performed with MDA-MB–231 and MDA-MB–157 lysates. (**A**) Cells treated for 24 h with 10 μM U0126 (lane 2) or dimethyl sulfoxide (DMSO) as a control (CTRL, lane 1). (**B**) MDA-MB–231 cells were treated with BI–D1870 and C646 (lane 2), and respective controls (CTRL, lane 1) and WB analyses were performed on cellular lysates. Representative red ponceau-stained membranes are shown as loading and quantification controls. Molecular weight markers (kDa) are indicated. All WB experiments were performed in triplicate, providing consistent results. Representative images are shown. (**C**) qRT-PCR analyses of MDA-MB-231 and MDA-MB-157 cells silenced for CBP (siCBP), RSK2 (siRSK2), or treated with a control small interfering RNA (siCTRL). The expression of HMGA1, RSK2, and CBP were analysed. Glyceraldehydes-3-phosphate dehydrogenase (GAPDH) was used as internal reference gene. Mean, standard deviations (*n* = 3), and statistical significance (*t*-test) are indicated (**: *p* < 0.01; ***: *p* < 0.001).

**Figure 8 cancers-11-01105-f008:**
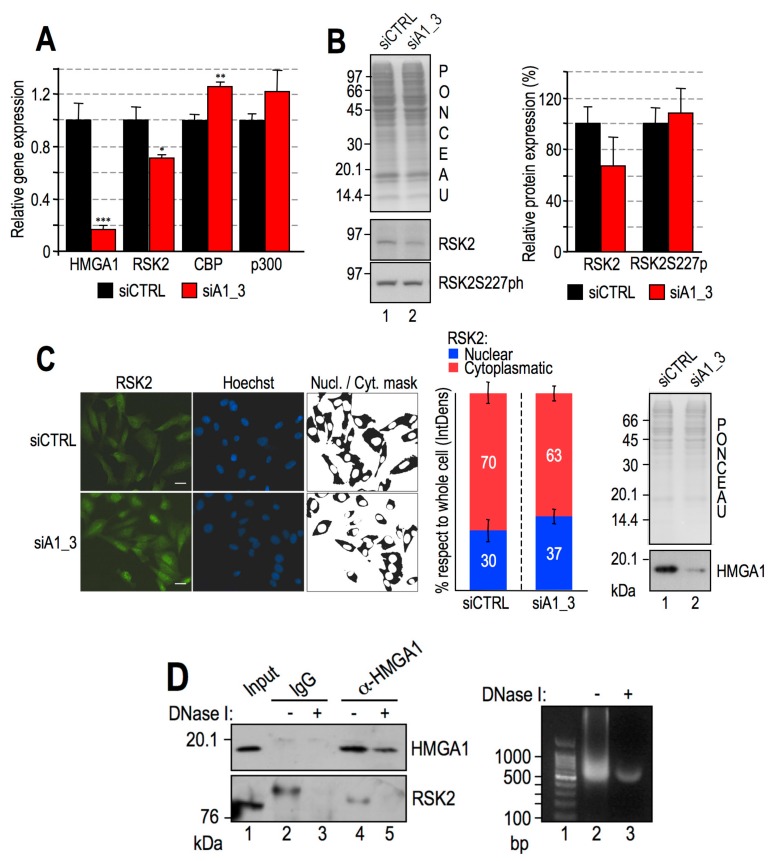
High Mobility Group A1 (HMGA1) modulates histone post-translational modifications (PTMs) acting downstream of Ribosomal protein S6 kinase alpha-3 (RSK2) phosphorylation. (**A**) Relative gene expression data regarding HMGA1, RSK2, CBP, and p300 obtained by reverse transcription-quantitative polymerase chain reaction (RT-qPCR) analyses of MDA-MB-231 samples treated with siCTRL or siA1_3. Mean and standard deviation (*n* = 3) are reported relative to the control sample. Glyceraldehydes-3-phosphate dehydrogenase (GAPDH) was used as an internal reference gene. Significance was assigned by Student’s *t*-test (*: *p* < 0.05; **: *p* < 0.01; ***: *p* < 0.001). (**B**) Western blot (WB) analysis of MDA-MB-231 lysates upon HMGA1 silencing (lane 2, siA1_3) or control (lane 1, siCTRL). Panels are representative of a biological triplicate, and the red ponceau-stained membrane is shown as a loading and quantification control. HMGA1 levels have been previously verified (see Figure 1). The histogram graph displays WB band densitometry values normalized to red ponceau-stained lysate lanes and relative to control samples. Mean and standard deviations (*n* = 3) are reported. Significance was assigned by Student’s *t*-test. (**C**) Immunofluorescence analysis performed with MDA-MB-231 cells treated with control siRNA (siCTRL) and with siRNA targeting HMGA1 (siA1_3). Representative fields are shown. Staining: RSK2 (green), DNA (Hoechst, blue). The nuclear/cytoplasm mask is reported. Results are shown as percentages of integrated densities (IntDens) of each area (nucleus or cytoplasm) with respect to the whole cell integrated density and reported as a mean of three different regions of interest for each sample type (siCTRL or siA1_3). The efficiency of HMGA1 silencing has been ascertained by WB and is shown on the right side. (**D**) HMGA1 and RSK2 co-immunoprecipitation (co–IP) analysis performed with MDA-MB-231 lysates. Input lysate was loaded in lane 1. Lanes 2 and 3 display immunoglobulins (IgG) immunoprecipitates (controls), while lanes 4 and 5 display HMGA1 immunoprecipitates. Lanes 3 and 5 show co–IPs performed with lysates treated with DNase I. WB analyses of HMGA1 and RSK2 are reported. The efficiency of DNase I digestion is shown on the right side.

**Figure 9 cancers-11-01105-f009:**
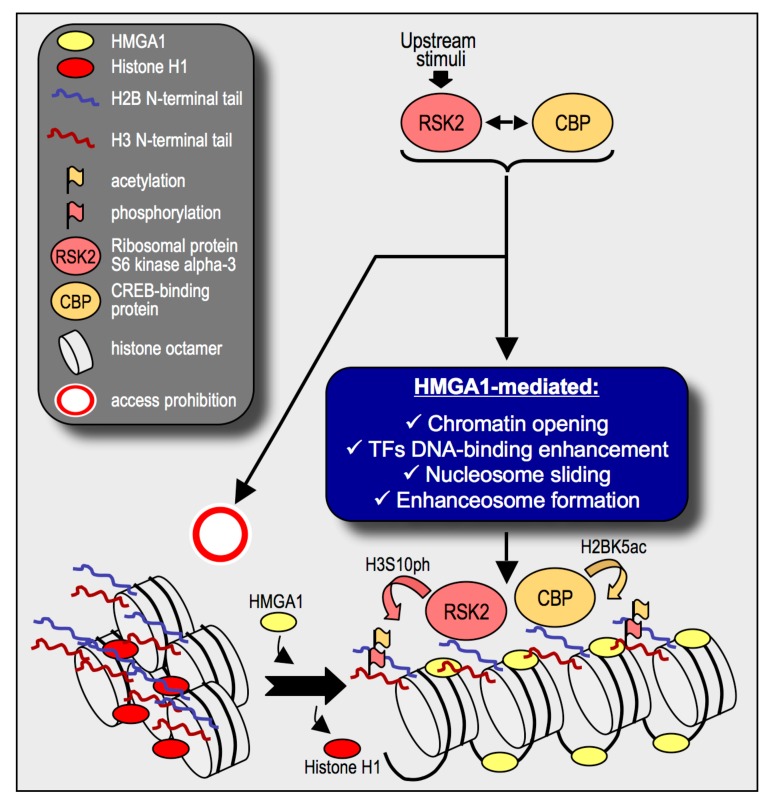
Expanding High Mobility Group A1 (HMGA1) chromatin functions. HMGA1 participates in the chromatin opening process by displacing histone H1 from chromatin via a competitive mechanism. Once bound to nucleosomes and/or DNA, HMGA1 is involved in the recruitment of transcription factors (TFs) and in the formation of protein platforms for the landing of other chromatin proteins [17,23]. In this work, we provided evidence that in triple negative breast cancer (TNBC) cells HMGA1 could be an essential co-factor for conveying onto chromatin ribosomal protein S6 kinase alpha-3 (RSK2) and CREB-binding protein (CBP) activities.

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
