# Peer review of "HMGA1 Modulates Gene Transcription Sustaining a Tumor Signalling Pathway Acting on the Epigenetic Status of Triple-Negative Breast Cancer Cells"

_cancers, 2019, doi:10.3390/cancers11081105_

Round 1
Reviewer 1 Report
Carlotta Penzo and colleagues reports the role of HMGA1 protein in modulating gene transcription by epigenetics of histone codes.
Minor:
Authors have cited multiple review articles. Some of the literature is not specific and citations need to some redoing on first paragraph of introduction sections. Additionally, paragraph 2 lines 51-57 need specific citations
Line 61 need rephrasing in context of disease specificity
Line 70 need specific examples of recent work and cited literature.
Fig.1 Figure can be divided in sub-pars A and B. Legends need modifications
Reference 31 should be replaced with original work
Lines 139, 140 need reference
Major:
It is noted that authors have not cited the original work from multiple laboratories. Authors have relied on review articles heavily. I would love to re-review this manuscript after authors cite and read original literature that has been used for forming hypothesis.
Additionally, authors have relied heavily on SI-RNA technique. Can some experiments be reproduce on stable HMGA1 knockout cells?
Reviewer 2 Report
The manuscript by Penzo et al represents a well composed and well presented manuscript exploring the role of HMG1A as modifier of chromatin accessibility and gene expression particularly in the previously reported HMG1A subset signature (Hss) of genes downstream of RAS signaling.
Strengths of this paper include the diversity of targets, assay and approaches to illustrate the importance of several genes (HMG1A, RSK2) in altering either gene expression or overall histone PTM (serine phosphorylation and methylation) in the MDA231 breast cancer cell line.
Major weakness in this paper are the use of only one cell line, however this is acceptable for this particular journal. The authors convincingly illustrate the importance of each of their key molecules (RSK2 pathway and HMG1A) in modifying gene expression of similar genes (HSSS) attempting to establish the flow of the signaling network (as diagramed in Figure 9) which would establish novel understandings previously unappreciated about these two molecules. However, the reviewer was confused in reconciling contradictions in interpretation of the data, that has confused the accuracy of the model they have proposed.
Of specific concern is in consistently and fairly interpreting the data. Figure 2C, they state that the impacts of siRNA against HMG1A DOES NOT significantly impact RSK2 (line 147). Yet in line 255 it is argue that HMG1A is necessary for RSK2 activities and that knockdown of HMG1A does impact RSK2 expression (line 272) While knocking down CBP phenocopies RSK2 and HMG1A, it is unclear, what impact if any knockdown of CBP/P300 have on RSK2 or HMG1A expression or if HMG1A or RSK2 knockdown or inhibtion impact CBP activity/binding (authors do show HMG1A knockdown does not impact CBP expression - what about localization). These stories require some connecting or clarifying thoughts or connecting data to validate the model they propose. As they authors have done an excellent job in eliminating concerns of MSK1/2 and P38MAPK, additional information needs to be provided other than phenocopying of gene expression in a limited set of genes to validate their model as the exclusive mechanism of action. While subject to the strength of interpretation of the observations depicted, their model mostly fits the data displayed, however are other models possible that also explain the data. If the proposed model is the strongest explain why, provide additional and clear details.
Minor Comments: - Unlike other figures - figure 1 does not report how stastical evaluations were performed (what test to acquire p values). The reviewer prefers to see individual dots of experimental repeats rather than only bar graphs but present form is acceptable.
The strength of figure 8 at this position of the manuscript is not apparent to the reviewer. This is another methodology that draws a connection between RSK2 and the phosphorylation of histones. Might serve better occuring before CBP story.
Overall a strong presentation of data from strong experiments. Some additional care in composition of the manuscript will more effectively and consistently communicate how the data fit their model, or what other models may explain their data or the gaps in their model not well explained by the presented data. I look forward to reading the revised manuscript and approving for publication.
Reviewer 3 Report
Title: HMGA1 Protein Affects Breast Cancer Cells Gene Transcription by Modulating the Histone Epigenetic Code
Manuscript ID: cancers-521015
Reviewer comments: The article describes a potential additional mechanism exploited by the High Mobility Group A 1 (HMGA1) non-histone protein to exert its oncogenic effect in a breast cancer cell model.
The article is interesting but a little ambiguous. Is this HMGA1 effect a general mechanism in all cells, in all cancer cells, specific sub-sets of cancer cells, exclusive to TNBC, or all breast cancer cells. Recommend major revision if it is to be accepted for publication.
Comment 1: The title is ambiguous, ‘Affects’ should be specified, ie promote BC aggressiveness??? Or promote tumor signalling pathways???
Comment 2: Abstract is too waffly, should be more succinct, direct and the point of the article.
Comment 3: Was the MDA-MB-468 a convenient cell line to use to demonstrate the mechanism of HMGA1 action because the idea of this being a mechanism of ‘oncogenic effect’ in TNBC seems to be secondary to the paper’s overall study. The study is more focused on the mechanism of HMGA1 in one cell line (MDA-MB-231) which happens to be a TNBC cell line.
- MDA-MB-468 is mentioned in the methods and the discussion but there are no results using these cell lines??
- Only one experiment shown with MDA-MD-157, the majority of the results only show MDA-MB-468? The authors should provide the results from all three cell lines to be inclusive.
- Following on from the general comment above - Is this a generic mechanism or is this mechanism exclusive to TNBC. It is very unclear if this is a unique mechanism to these cell lines or all breast cancers or even all cancers or normal cell lines. There is no information or comparison - the title is very specific to breast cancer - has the authors looked at any other cell lines. Is there any information in human breast tissue?
Comment 4: Can the authors comment or discuss the mechanism described - exclusively affecting only one pathway (RAS) or is it a master regulator of other pathways?
Comment 5: The authors states ‘Finally, we show that these two effectors are involved in the gene expression regulation of factors involved in cancer aggressiveness.’ The wound healing assays are inconclusive and there is no functional endpoint shown to how this mechanism affects the cancer biology of the cell ie proliferation, or migration. Perhaps the authors should do some cell proliferation assays or simple counts or migration assays and some more convincing images of changes in morphology.
Comment 6: Clarity of introduction…. The first sentence reads, " Most of the stimuli a cell is subjected to..."? explain. Also throughout the manuscript past and present tenses are misused (the authors used present and past tenses randomly). For example, line 73"We investigated..", line 75"We provided..", line 78"We show...". Same problem can be seen in the Materials and Methods section. The authors used "we suggested" in many occasions in the paper. A better expression should " The data suggested, or the results suggested".
Comment 7: ‘HMGA1 Expression Influences the Histone Code’ explain what you mean in this section. No reference to histone code in this section – need to link the concept to the explanation in the results.
Comment 8: Results are presented in the introduction, repeated in results (suggest remove).
Comment 8: When abbreviating in the body of the text for the first time (excluding abstract) should be full name ie line 51 HMGA.
Comment 9: Figure 4a need clearer images. Wound healing assays are not convincing (4b). The visual results do not match the analyses (4b and c).
Comment 10: Please explain in the figure legend or the methods - nuclear/cytoplasmic mask.
Round 2
Reviewer 1 Report
Authors have made appropriate changes. I have no further suggestions on the manuscript.